# Aggregates Associated with Instability of Antibodies during Aerosolization Induce Adverse Immunological Effects

**DOI:** 10.3390/pharmaceutics14030671

**Published:** 2022-03-18

**Authors:** Thomas Sécher, Elsa Bodier-Montagutelli, Christelle Parent, Laura Bouvart, Mélanie Cortes, Marion Ferreira, Ronan MacLoughlin, Guy Ilango, Otmar Schmid, Renaud Respaud, Nathalie Heuzé-Vourc’h

**Affiliations:** 1INSERM, Centre d’Etude des Pathologies Respiratoires, U1100, F-37032 Tours, France; secher.thomas@gmail.com (T.S.); montage@ch-blois.fr (E.B.-M.); christelle.parent@univ-tours.fr (C.P.); laura.bouvart@gmail.com (L.B.); cortes.melanie@wanadoo.fr (M.C.); marion-ferreira@hotmail.fr (M.F.); guy.ilango@univ-tours.fr (G.I.); renaud.respaud@gmail.com (R.R.); 2Faculté de Médecine, Université de Tours, F-37032 Tours, France; 3Service de Pharmacie, Centre Hospitalier Régional Universitaire de Tours, F-37032 Tours, France; 4Département de Médecine Pédiatrique, Centre Hospitalier Régional Universitaire de Tours, F-37032 Tours, France; 5Département de Pneumologie et d’Exploration Respiratoire Fonctionnelle, Centre Hospitalier Régional Universitaire de Tours, F-37032 Tours, France; 6Research and Development, Science and Emerging Technologies, Aerogen Limited, Galway Business Park, H91 HE94 Galway, Ireland; rmacloughlin@aerogen.com; 7School of Pharmacy & Biomolecular Sciences, Royal College of Surgeons in Ireland, D02 YN77 Dublin, Ireland; 8School of Pharmacy and Pharmaceutical Sciences, Trinity College, D02 PN40 Dublin, Ireland; 9Institute of Lung Health and Immunology/Comprehensive Pneumology Center with the CPC-M bioArchive, Helmholtz Zentrum München, Member of the German Center for Lung Research (DZL), 85764 Munich, Germany; otmar.schmid@helmholtz-muenchen.de

**Keywords:** therapeutic antibody, aerosol, aggregates, immunogenicity

## Abstract

Background: Immunogenicity refers to the inherent ability of a molecule to stimulate an immune response. Aggregates are one of the major risk factors for the undesired immunogenicity of therapeutic antibodies (Ab) and may ultimately result in immune-mediated adverse effects. For Ab delivered by inhalation, it is necessary to consider the interaction between aggregates resulting from the instability of the Ab during aerosolization and the lung mucosa. The aim of this study was to determine the impact of aggregates produced during aerosolization of therapeutic Ab on the immune system. Methods: Human and murine immunoglobulin G (IgG) were aerosolized using a clinically-relevant nebulizer and their immunogenic potency was assessed, both in vitro using a standard human monocyte-derived dendritic cell (MoDC) reporter assay and in vivo in immune cells in the airway compartment, lung parenchyma and spleen of healthy C57BL/6 mice after pulmonary administration. Results: IgG aggregates, produced during nebulization, induced a dose-dependent activation of MoDC characterized by the enhanced production of cytokines and expression of co-stimulatory markers. Interestingly, in vivo administration of high amounts of nebulization-mediated IgG aggregates resulted in a profound and sustained local and systemic depletion of immune cells, which was attributable to cell death. This cytotoxic effect was observed when nebulized IgG was administered locally in the airways as compared to a systemic administration but was mitigated by improving IgG stability during nebulization, through the addition of polysorbates to the formulation. Conclusion: Although inhalation delivery represents an attractive alternative route for delivering Ab to treat respiratory infections, our findings indicate that it is critical to prevent IgG aggregation during the nebulization process to avoid pro-inflammatory and cytotoxic effects. The optimization of Ab formulation can mitigate adverse effects induced by nebulization.

## 1. Introduction

Therapeutic antibodies (Ab), which mainly consist of monoclonal IgG, represent the fastest growing class of protein therapeutics, accounting for 90% of proteins on the market [1]. As recently highlighted during the COVID-19 pandemic, Ab have a tremendous potential to provide a rapid neutralizing response to emerging viral respiratory pathogens, augmenting vaccines for the control of respiratory infections. Interestingly, both pulmonary and nasal delivery of therapeutic proteins are receiving increasing interest, as the airways are a relevant non-invasive entry portal to the respiratory tract for local-acting protein therapeutics [2,3,4]. For the treatment of respiratory diseases, several preclinical studies have demonstrated that inhaled protein therapeutics, including Ab, are efficacious and display a better pharmacokinetic profile, as compared to other routes [5,6,7]. Despite the recent clinical development of oral anti-infective Ab, in the context of the global response to SARS-CoV2 pandemic, the inhalation route remains unexploited for Ab. One of the unknowns is the biological consequence of Ab instability during aerosolization. Inhalation requires transformation of a (bulk) protein formulation in an aerosol, i.e., dispersion of a solution/suspension or a dry-powder into micron-sized particles suspended in a gaseous medium. As with 75% of the inhaled protein therapeutics in clinical development, nebulization of liquid formulations is often the primary technique used for the inhalation of proteins [5]. Nebulization generates a significant air–liquid interface which, combined with the potential for nebulization-induced temperature increase and/or shear forces, can be deleterious for proteins. In response to such stresses, proteins are prone to unfolding, aggregating and, in some cases, being partly inactivated [8,9]. Aggregation is a key marker of instability in full-length Ab during nebulization [10,11] and its extent mainly depends on the type of Ab, the aerosol generator type and the formulation characteristics.

Aggregates are associated with Ab-related adverse immunogenicity [12,13,14]. Immunogenicity refers to the inherent properties of a molecule to stimulate an immune response. Adverse immunogenicity is due to uncontrolled and protracted immune responses and has major consequences for product safety and pharmacology [15,16,17]. For instance, adverse immunogenicity is associated with patient’s immunization and production of anti-drug antibodies (ADAs) and affects protein therapeutic pharmacokinetics (PK), pharmacodynamics, efficacy and safety, sometimes resulting in extremely harmful side effects [18,19]. To date, many factors have been implicated in adverse immunogenicity [20,21,22,23] and they include the content of aggregates and the route of administration, which are important parameters for Ab inhalation.

After inhalation, aggregates from aerosolized Ab will encounter the airways immune system, which has evolved over time to recognize and prevent foreign particles, including aggregates, from penetrating the body. The airway mucosa is sentinelled by a high density of antigen-presenting cells (APC) that quickly and efficiently orchestrate local immune responses against inhaled antigens [24]. In the present study, we analyzed, both in vitro and in vivo, the immunological consequences of Ab aggregates generated during aerosolization and delivered through the airways. First, we screened the potency of IgG aggregates generated by nebulization to activate APC in vitro, using a standard human monocyte-derived dendritic cell (MoDC) assay. Considering the complexity of the immune system and the lung mucosal environment, we evaluated the impact of nebulization-mediated IgG aggregates on immune cells in vivo after pulmonary administration. Our findings show that aggregation attributable to Ab nebulization induced immune cell activation, in a dose-dependent manner and delivering a high-level of aggregates through the pulmonary route had a dramatic effect on immune cell homeostasis, but also that this was avoidable through appropriate formulation approaches.

## 2. Materials and Methods

Mice

Adult male C57BL/6jrj (B6) mice (5 to 7 weeks old) were obtained from Janvier (France). All mice were housed under specific pathogen-free conditions at the PST Animaleries animal facility (France) and had access to food and water ad libitum. All animal experiments complied with the current European legislative, regulatory and ethical requirements and were approved by the local animal care and use committee (reference: APAFIS No.10200-2017061311352787).

Antibodies

Abs 1, 2 and 3 are full-length IgG1 (named hIgG1-1, hIgG1-2 and hIgG1-3) used in the clinics, supplied in their commercial formulation with endotoxin levels meeting acceptance thresholds (according to their certificate of analysis). To avoid any interference on the aggregation propensity of each Ab, excipients were removed by hydroxyapatite chromatography and subsequent dialysis against phosphate-buffered saline (1X-PBS). Protein concentration was then evaluated for each Ab: hIgG1-1 (1.7 mg/mL), hIgG1-2 (1.9 mg/mL) and hIgG1-3 (2.38 mg/mL)

mAb166 (mIgG2b-1) is a murine monoclonal IgG2b,κ Ab against pcrV, a component of a type three secretion system (T3SS) of *Pseudomonas aeruginosa* [25]. MPC11 (mIgG2b-2) is the control isotype of mAb166. Both were supplied as sterile, pyrogen-free solution in 1X-phosphate-buffered saline (1X-1X-PBS), in accordance with good manufacturing practice, by BioXcell (New Lebanon, NH, USA) with endotoxin level < 1 EU/mL (according to their certificate of analysis). They were supplied as follows: mIgG2b-1 (2.5 mg/mL), mIgG2b-2 (8.5 mg/mL). In some experiments, Ab formulations were supplemented with Polysorbate 80 (PS80; Sigma–Aldrich, Saint-Quentin Fallavier, France) at 0.01% or 0.05% (final volume) prior to nebulization.

Antibody nebulization

For sterility purposes, this procedure was performed under a cell-culture hood. All antibodies were filtered in suspension on a 0.22 µm syringe filter (Millipore, Guyancourt, France). For in vitro assay, all antibodies were diluted in PBS1X to a final concentration of ~1.7 mg/mL before nebulization. For each Ab, 1 mL were nebulized using the clinically-relevant Aeroneb Pro^TM^ vibrating-mesh nebulizer (Aerogen, Ireland), connected with a 13 mL polypropylene tube (Dutscher, Bernolsheim, France) or the VITROCELL Cloud 12 system (VITROCELL Systems, Waldkirch, Germany), which is the commercial version of the Air–Liquid Interface Cell Exposure–Cloud (ALICE-cloud) system, described by Lenz and colleagues [26]. Nebulization duration was measured and lasted ~5 min for each sample (i.e., 0.4 mL/min liquid output rate). For in vitro aerosol-cell exposures, using the VITROCELL Cloud 12 system, the aerosol cloud was allowed to settle for 20 min before removing cells. When necessary, nebulized Ab solutions were filtered on a 0.45 µm syringe filter (Millipore, Guyancourt, France) prior to further use. Nebulizers were washed extensively between each nebulization sessions using 0.22 µm autoclaved water and 1X-PBS. Subsequently, nebulized 1X-PBS was analyzed by flow cell microscopy and the nebulizer was considered cleaned and operating for Ab nebulization when the nebulized 1X-PBS samples contained less than 300 particles/mL. To rule out that cell activation may be attributable to contamination released during nebulization, the activation potency of nebulized 1X-PBS (vehicle solution) was also investigated in vitro. Our analysis revealed that nebulized 1X-PBS had no impact on MoDC activation (data not shown).

Dynamic Light Scattering (DLS) analysis of antibodies

Native and nebulized Ab were analyzed by dynamic light scattering with a Dynapro Nanostar^®^ (Wyatt Technology, Goleta, CA, USA) after appropriate dilution to 100 μg/mL in 1X-PBS in UVette (Eppendorf, Montesson, France). For each sample, the acquisition was performed 10 times for 7 s each, at a temperature of 25 °C with a detection angle of 90°. The particle diameter (nm) and polydispersion index (pdi) were analyzed using DynamicsTM software (Wyatt Technology, Goleta, CA, USA). Samples with less than 70% of successful analyses were considered multimodal and non-analyzable as recommended by the manufacturer (depicted as n/a).

Flow cell microscopy (FCM) of antibodies

Native and nebulized Ab were analyzed by flow cell microscopy with an Occhio^®^ FC200S+ (Occhio, Angleur, Belgium) after appropriate dilution to 100 μg/mL in medium in the flowcell. Briefly, 250 µL of each antibody solution passed continuously in the flow cell, where particles were automatically detected, sized and counted by a camera. Particle counting, size assessment and distribution were analyzed using CallistoTM software (Occhio, Angleur, Belgium).

Monocyte-derived dendritic cells (MoDC) preparation

Human peripheral mononuclear cells (PBMC) were purified from cytapheresis, obtained from naïve donors (Etablissement Français du Sang, Centre Hospitalier Régional Universitaire Bretonneau, Tours, France) by centrifugation on a Ficoll density gradient (Eurobio, Les Ulis, France). MoDC were prepared from PMBC as previously described [27]. Briefly, CD14+ monocytes were isolated using magnetic cell sorting (Miltenyi Biotec, Bergisch Gladbach, Germany) and cultured during 6 days, at 1 × 10^6^ cells/mL in RPMI1640-GlutamaxTM (Gibco, Illkirch, France) supplemented with 10% of FCS (Dutscher, France), 1X-Penicillin/Streptomycin (Gibco, Illkirch, France) in the presence of 25 ng/mL of IL-4 (Miltenyi Biotec, Bergisch Gladbach, Germany) and 100 ng/mL of GM-CSF. On day 6, homogeneity and viability of MoDC population was checked by flow cytometry based on their physical characteristics (forward scatter (FSC) and side scatter (SSC)) and incorporation of vital dye (LiveDead^TM^, Invitrogen, Illkirch, France). The immature phenotype of MoDC was checked based on their DC-SIGN+, CD80low, CD83low and HLA-DR low phenotype.

MoDC stimulation with antibody preparation

On day 6, immature MoDC were harvested and washed in complete medium (RPMI1640-Glutamax™ supplemented with 10% of FCS (Dutscher, Bernolsheim, France) and 1X-Penicillin/Streptomycin). For experiments using Ab aerosols collected in a 13 mL polypropylene tube, 1 × 10^5^ cells/well were plated in 75 µL in a U-bottom 96-well plate (Falcon, Becton Dickinson, France) for 4 h. A total of 75 µL of nebulized or native Ab was then added to a final concentration of 1, 10, 100 and 200 µg/mL depending on the experiments. For experiments using the VITROCELL Cloud 12 system, 1 × 10^5^ cells/well were plated in 150 µL on a 24-well plate permeable insert (Corning, Hazebrouck, France) for 18 h. The insert was then placed in the VITROCELL Cloud 12 system with 200 µL in the basolateral compartment to prevent cell drying and cells were exposed to Ab aerosol as described above. Inserts were then put back on a companion 24-well plate.

Cells were incubated in 150 µL of complete medium for 18 h at 37 °C with 5% CO_2_. Lipopolysaccharide (LPS, from *Escherichia coli* O111:B5, Sigma-Aldrich, France) was used at 1 µg/mL as a positive control. Untreated and nebulized -1X-PBS (Gibco, Illkirch, France) were used as vehicle controls. Each condition was tested in 3–9 replicates.

Analysis of MoDC activation by flow cytometry

After 18 h of stimulation, MoDC were harvested and saturated in -1X-PBS supplemented with 2% of FCS, 2 mM EDTA and 1X-human Fc-Block (Becton Dickinson, France) for 15 min at 4 °C. Cells were washed and stained in FACS buffer (1X-PBS supplemented with 2% FCS and 2 mM EDTA) with antibodies described in Appendix A, for 20 min at 4 °C in the dark. FMO controls were generated for each parameter analyzed. All antibodies were from Biolegend (London, UK). Data acquisition was conducted on the 8-colors MACSQuant flow cytometer (Miltenyi Biotec, Bergisch Gladbach, Germany) on a minimum of 10,000 living cells. MoDC analysis was performed using VenturiOne software (Applied Cytometry, Sheffield, UK). Analysis was conducted on singlet (FSC-A/FSC-H gating) and CD45+ live cells (negative for LiveDead staining). Costimulatory protein expression was expressed as a ratio of the median fluorescence intensity (MFI) of cells treated with nebulized Ab/MFI of cells treated with native Ab.

Cytokine, chemokine and protein assays

All assays were performed on Immulon^TM^ 96-well plates (ThermoFischer Scientific, Illkirch, France). Concentrations of CXCL8 (IL-8), human IL-6 in cell-free MoDC supernatants and TNF, IL6, IL1b and CXCL1 (KC) were measured using specific ELISAs (Biolegend, London, UK; limit of detection: at 15.6 pg/mL) according to the manufacturer’s instructions and were normalized based on the mean concentration of total protein. Total protein concentration was determined using a BCA assay (ThermoFischer Scientific, Illkirch, France), according to the manufacturer’s instructions (limit of detection: 15 µg/mL). Protein expression was expressed as a ratio of the protein concentration of cells treated with nebulized Ab/protein concentration of cells treated with native Ab.

Animal experiments

For each experiment, 5 mice per group were used. Animal experiments were performed 2–3 times. Nebulized or native mIgG2b-1 and mIgG2b-2 (100 µg/animal, in 40 µL) were administered orotracheally or through intravenous injection using a restraining tube at days 0, 7, 14, 21 and 28. For orotracheal administration, mice were anesthetized with isofluorane 4% and an operating otoscope fit with intubation specula was introduced both to maintain tongue retraction and to visualize the glottis. A fiber optic wire threaded through a 20 G catheter and connected to torch stylet (Harvard Apparatus, Holliston, MA, USA) was inserted into the mouse trachea. Correct intubation was confirmed using a lung inflation bulb test and 40 µL of the bacterial solution was applied using an ultrafine pipette tip. For the acute treatment assessment, animals were euthanized using a lethal dose of ketamine/xylazine at 4 h, on day 1 and 14 days after a single administration. In the chronic treatment groups, animals were euthanized at day 29.

At the time of necropsy, blood was recovered by intracardiac puncture. Bronchoalveolar lavage fluid (BALF) was then collected by cannulating the trachea and washing the lung twice with 1.2 mL of 1X-PBS at room temperature. The lavage fluid was centrifuged at 400× *g* for 10 min at 4 °C, and the supernatant was stored at −20 °C until analysis. The cell pellet was resuspended in FACS buffer and counted in a hemocytometer chamber. Peripheral blood was washed out by intracardiac perfusion with 10 mL of 1X-PBS. Lung and spleen homogenates were then prepared in 2 mL of RPMI1640 containing 125 µg/mL of Liberase (Sigma–Aldrich, France) and 100 µg/mL of DnaseI (Sigma–Aldrich, Saint-Quentin Fallavier, France) using a GentleMACS tissue homogenizer (Miltenyi Biotec, Bergisch Gladbach, Germany). Cells were isolated through a 100 µm cell strainer and purified with a 20% Percoll (Sigma–Aldrich, Saint-Quentin Fallavier, France) density gradient. Cell preparation was centrifuged at 400× *g* for 10 min at 4 °C, and the pellet was resuspended in FACS buffer, and counted in a hemocytometer chamber.

Immune cell phenotyping by flow cytometry

BALF, lung and spleen cells isolated as described above were saturated in 1X-PBS supplemented with 2% of FCS, 2 mM EDTA and 1X- human Fc-Block (Becton Dickinson, France) for 15 min at 4 °C. Cells were washed and stained with antibodies described in Appendix A for 20 min at 4 °C in the dark. Data acquisition was made on the 8-colors MACSQuant flow cytometer (Miltenyi Biotec, Bergisch Gladbach, Germany) on a minimum of 100,000 living cells. Analysis was performed using VenturiOne software (Applied Cytometry, Sheffield, UK). Analysis was made on singlet (FSC-A/FSC-H gating) and CD45+ live cells (negative for LiveDead staining). Immune cell phenotypes were determined as described in Appendix A. Data were normalized based on the mean concentration of respective native Ab conditions or expressed as a number of cells calculated as follows: % of the total CD45+ immune cell population x total number of cells. For Annexin-V/PI staining, data were normalized as follows: % of the positive population for nebulized Ab/% of the positive population for native Ab.

Histology

Lungs were fixed in 10% buffered formalin (Shandon), dehydrated in ethanol and embedded in paraffin. Serial sections (3 mm) were stained with hematoxylin and eosin (HE) or Congo red. Ten HE sections per mouse were randomly evaluated to count the cell nucleus using a machine-learning pixel classification plugin for Image J (https://imagej.net/plugins/tws/, as available on 17 March 2022), as described previously [28].

Statistical analysis

Differences between experimental groups were determined using Kruskal –Wallis or t-test comparing two groups, using one or two-way analysis of variance (ANOVA) followed by Newman–Keuls or Bonferroni post-test (for comparison between more than two groups), after confirmation that the data were normally distributed. All statistical tests were performed with GraphPad Prism software (version 4.03 for Windows, GraphPad Software Inc., San Diego, CA, USA). All data are presented as mean ± standard error of the mean (SEM). The threshold for statistical significance was set to *p* < 0.05.

## 3. Results

### 3.1. Aggregation of Antibodies Is Heterogeneous during Nebulization

Nebulization promotes Ab aggregation [10,11]. In this study, mesh nebulization was used since it is less deleterious on Ab and often considered for aerosolization of proteins [5]. Full-length Ab, commercially available or under preclinical development, were reformulated in 1X-PBS and nebulized with the Aeroneb Pro™ vibrating-mesh nebulizer.

Ab aggregation results in a broad range of particles, from dimers (several nanometers) to micron-sized, and even visible particles in some cases [29], requiring the combination of different complementary analytical techniques. The aggregation profiles of Ab were assessed using DLS and FCM to report particles at the submicrometric and micrometric scales, respectively (Figure 1).

Nebulization led to an increase in the concentration of particles > 2 µm for hIgG1-1, mIgG2b-1 and mIgG2b-2, as compared to their native counterparts. Conversely, no significant increase in particle count was observed for hIgG1-2 and hIgG1-3 (Figure 1). Furthermore, comparing nebulized Ab, we observed that hIgG1-1 and mIgG2b-1 had significantly more particles than the other human and murine Ab, respectively. In addition to differences in total particle concentration, there were also differences in particle size distribution. Nebulized hIgG1-1 and -2 had more than 78% of their particles smaller than 5 µm, while hIgG1-3, mIgG2b-1 and mIgG2b-2 comprised more than 33% of their particles above 5 µm (Table 1). For murine mIgG2b-1 and mIgG2b-2, aggregation was also observed at the submicron scale, most notably for nebulized mIgG2b-1, which could not even be analyzed by DLS due to the large heterogeneity of the particle size distribution (most likely all-sized Ab aggregates), which cannot be clearly structured in defined size modes—a prerequisite for DLS measurements. Nebulized mIgG2b-2 exhibited a significant reduction (7%) in monomeric Ab amount (Appendix A). Overall, our results highlight the substantial heterogeneity of Ab aggregation during mesh nebulization, most likely depending on Ab sequence/structure.

### 3.2. Ab Aggregates, Produced during Mesh Nebulization, Activate Antigen-Presenting Cells

The dramatic consequences of adverse immunogenicity have prompted regulatory authorities to establish a guidance to test the immunogenicity of therapeutic protein products and industry to propose screening approaches [30,31]. They include investigating the ability of protein aggregates to activate immune cells, in vitro.

Here, we incubated human monocyte-derived dendritic cells (MoDC) overnight with native or nebulized Abs (hIgG1-1 to -3) and analyzed both the release of pro-inflammatory cytokines and expression of co-stimulatory proteins involved in the DC-T synapse (CD25, CD83, CD86 and CD80). Neither the native Ab, nor the nebulized buffer without Ab (data not shown) promoted IL-6 or IL-8 production by MoDC (Figure 2A,B) and modulated cell markers as compared to untreated MoDC (Figure 2E–H). Nebulized and aggregated hIgG1-1 induced a significant and dose-dependent increase in cytokine production (Figure 2A–D) whereas the other nebulized Ab solutions had only a minor and inconsistent impact on cytokine level. The nebulized and aggregated hIgG1-1 induced a slight but significant increase in all cell markers, whereas the effect of other nebulized Ab was limited (Figure 2E–H). Interestingly, the filtration of Ab solutions after nebulization, removing micrometric particles (data not shown), resulted in the abrogation of both cytokine release and expression of co-stimulatory markers by MoDC (Figure 2A–H, gray bars). In addition, when comparing nebulized antibodies, our analysis revealed that activation potency of hIgG1-1 was higher than for hIgG1-2 and hIgG1-3. Altogether, our results suggest that activation of APC was attributable to the presence of aggregates and that the extent of this activation was correlated to the number of particles (Figure 1). Since the aerosol collection system may modulate the amount and the size distribution of Ab aggregates after nebulization [11], we determined whether collection might induce a bias in APC activation. We directly exposed MoDC to hIgG1-1 aerosols using the VITROCELL Cloud 12 system [26]. As observed in Appendix A, we obtained similar results on MoDC activation independent of whether the cells were directly exposed to hIgG1-1 aerosols (through use of the VITROCELL Cloud 12 system) or exposed to them after collection as a bulk solution, supporting our decision of adopting the latter approach. Collectively, our results showed that nebulized Ab induced MoDC maturation and activation as compared to native Ab and confirmed the involvement of Ab aggregates in this response.

### 3.3. High-Level of Nebulization-Mediated Antibody Aggregates Impair Lung Cell Homeostasis after Lung Delivery

Historically, in vitro assays have been widely used to describe the potential immunogenicity of biotherapeutic aggregates [13]. However, they display several limitations: (i) the amount of aggregates inducing a response in vitro may not directly translate into in vivo response and (ii) they may not predict the impact of the pulmonary delivery route. To gain insight into the broad effects of Ab aggregates produced by aerosolization within the lung compartment, we selected two murine IgG2b–mIgG2b-1 and mIgG2b-2-producing different amounts of aggregates during mesh nebulization (Figure 1) which differentially activate MoDC (Appendix A). We administered them through the airways in naive mice. Remarkably, the nebulized and aggregated mIgG2b-1 resulted in a dramatic reduction in the total cell number in the airway compartment of mice (BAL), as compared to native antibodies administered by the same route (Figure 3A). Moreover, nebulized mIgG2b-2, producing 10-fold fewer aggregates after mesh nebulization than mIgG2b-1 (Figure 1), showed no statistically significant reduction in BAL cell number (Figure 3G). Of note, the animals that received the native Ab through the airways had similar cell count as sham animals (data not shown), which implies that the orotracheal application itself did not obfuscate our results.

Analysis of the cellular phenotype of the BAL cells revealed an analogous result for immune (CD45+ leukocytes) cells (Figure 3B,H). Further analysis did not reveal any impact on specific lineage as almost all immune cell types were affected after the administration of aggregated Ab (Appendix A). In the lung tissue, nebulized mIgG2b-1 caused a 2-fold reduction in total cell and leukocyte counts relative to controls (Figure 3C,D), while nebulization of mIgG2b-2 did not modify leukocyte counts (Figure 3I,J). This decrease affected both myeloid (neutrophils, monocytes/macrophages and dendritic cells) (Appendix A) and lymphoid lineages (B, T CD4+/CD8+ cells) (Appendix A). The modification of cell homeostasis in the airway compartment was not associated with an alteration of the lung epithelial barrier (Appendix A), but was mostly attributable to micron-sized particles as 0.45 µm-filtration of nebulized mIgG2b-1 prevented these adverse effects (Figure 3A–F, gray bars). The filtration step did not significantly modify Ab particle size distribution (Appendix A) or concentration (Appendix A), as compared to native Ab.

Interestingly, we observed protein aggregates (appearing as apple-green birefringence structures under polarized light) on Congo red stained lung sections from animals treated with nebulized mIgG2b-1 (Figure 3L). Unsupervised machine learning, which was used to quantify the cell nucleus on HE-stained lung section, confirmed that local administration of nebulized mIgG2b-1 was associated with a significant reduction of lung cells after 18 h (Figure 3K,M). Unexpectedly, the number of total and immune cells was also diminished, even though to a lesser extent, in the spleen (Figure 3E,F), indicating that the impairment of cellular homeostasis reached the systemic compartment. Contraction of cell number 18 h after a single airway administration of nebulized Ab primarily occurred in the airways, and then probably extended through the lung tissue and systemically as it was restricted to the airway compartment after 4 h (Figure 4A–F). Moreover, this effect was sustained for at least up to 14 days after Ab administration (Figure 4G,H), or after repeated administrations (Figure 4I–N). For either single or repeated administrations of nebulized mIg2b-1, we did not observe any sign of general toxicity, including body-weight loss (data not shown). Overall, our results suggest that airway administration of aggregated IgG (>0.45 µm) profoundly affected cellular homeostasis, in a time-dependent manner, both locally and systemically.

### 3.4. Nebulized Aggregated Antibody Induced Immunologically Silent Cell Death after Lung Administration

Next, we investigated the mechanisms accounting for host cell contraction and hypothesized that it was associated with cell death. Cell death occurs in multiple forms and can be divided in accidental cell death (ACD; necrosis) or regulated cell death (RCD; apoptosis) [32]. ACD is characterized by a dramatic and instantaneous collapse of cells and can be triggered in response to different stresses, including chemical, physical or mechanical insults, whereas RCD relies on committed molecular machinery [33]. Using Annexin-V/propidium iodide (PI) staining, which allows the discrimination of early apoptotic cells (Annexin-V+/PI-), late apoptotic cells (Annexin-V+/PI+) and necrotic cells (Annexin-V-/PI+) [34], we quantified the proportion of each cell death phenotypes in both total cells and CD45+ leukocytes population, 18 h after a single administration of mIgG2b-1. We observed that administration of nebulized mIgG2b-1 provoked a significant increase in both late apoptotic and necrotic spleen cells while lung and airway cells were suffering from necrosis as compared to mice treated with native Ab (Figure 5A–F) or with nebulized mIgG2b-2 (Appendix A). The type of cell death could also be determined by the analysis of mediators released in the environment. When comparing BALs from animals treated with native and nebulized mIgG2b-1, we did not observe any significant difference in the production of TNF, IL-6, IL-1b or CXCL1 (KC) (Figure 5G–J). These data suggest that the cell contraction occurring after single or multiple airway administration of nebulized and aggregated IgG was associated with an inflammatory silent cell death process.

### 3.5. The Effect of Nebulization-Mediated Antibody Aggregates on Immune Cell Homeostasis Is Specific of the Pulmonary Route

Immunogenicity of Ab is also dependent on their route of administration [20,21,22]. Thus, we investigated the effect of native or nebulized-mIgG2b-1 after intravenous injection. In contrast to what was observed after airway administration, there were no significant differences in the BAL or lung cell counts in the animals who received native or nebulized mIgG2b-1 intravenously (Figure 6A–D). Interestingly, these results were substantiated when analyzing cell number after repeated administration of nebulized mIgG2b-1 (Appendix A), where no differences were noticed in the airways or lungs of animals treated by repeated intravenous injections as compared to animals, which received the IgG in the lungs. Our data suggest that the route of administration played an important role on the adverse effect of IgG aggregates produced during nebulization on cell homeostasis.

### 3.6. Reducing Aggregation Limits Pulmonary Cytotoxicity Associated to Lung Administration of Nebulized Ab

Pharmaceutical development aims to design a high-quality product ensuring an efficacious and safe treatment along the life of the product. Hence, formulation and protein engineering are often adapted to limit Ab aggregation, especially considering chronic-based therapies. Different parameters, including addition of surfactant, have a protective effect, limiting Ab aggregation during nebulization [10]. Here, we added polysorbate 80 (PS80) in mIgG2b-1 formulation which significantly reduced its aggregation during nebulization (Appendix A). The addition of surfactant did not significantly modify Ab particle size distribution (Appendix A) or concentration (Appendix A) as compared to unformulated Ab. Single administration of nebulized mIgG2b-1 supplemented with 0.05% of PS80 abrogated the reduction of lung cell number (Figure 7C,D) and to a lesser extent of airway cell (Figure 7A,B) as compared to non-formulated mIgG2b-1. This was likely attributable to a reduction in cell death in the same compartment (Appendix A). These results suggest that optimizing IgG formulation improved its molecular stability and might limit adverse effects on cell homeostasis.

## 4. Discussion

Ab are highly sensitive to stresses encountered during their product development, manufacturing, storage or clinical use, and they often require high therapeutic doses, necessitating high concentration drug products with a higher risk of aggregation. Tracking aggregates prior and after Ab bioprocessing is essential to avoid risks for patients, treatment failure and ultimately termination of drug development/commercialization.

Protein aggregation has been widely demonstrated as an influential factor in the induction of adverse immunogenicity of biotherapeutics [35,36]. However, some of the stress conditions used in the literature are not representative of those experienced during product development, manufacturing, storage or clinical use, making the generated aggregates far different from those found in marketed products. This may hamper conclusions about the exact potency of protein aggregation in the occurrence of drug immunogenicity. Several intrinsic or extrinsic attributes of aggregates might work synergistically to induce immunogenicity. Among them, parameters related to the delivery route can affect the immune response associated with the administration of drugs. For example, subcutaneous delivery has been associated with higher immunogenicity than the intravenous route, for several biotherapeutics [37,38]. To our knowledge, there has not been any study conducted so far to assess the immunogenicity associated with lung administration of an aerosolized and aggregated Ab. We set out a dual experimental approach to investigate the quality and the extent of the immune responses induced by a nebulized and aggregated Ab using both in vitro and in vivo models. Here, we showed for the first time that the aggregates resulting from IgG nebulization induced immune cell over-activation and that their delivery to the lung markedly and durably impaired cell homeostasis.

Protein aggregation is a process of non-specific association of monomers through multiple physical and chemical pathways, which are well documented [39]. The characteristics of protein aggregates are variable in terms of particle size, number, morphology, chemical modifications, reversibility, conformation or hydrophobicity [40]. This phenotypic heterogeneity results from the various stresses applied to proteins and require specific analytical methods [41]. Aerosol generation involves the dispersion of liquid droplets into a gas. This process may be associated with physical stresses, including temperature variations and the generation of a massive air–liquid interface, which ultimately induce changes in protein conformation and lead to its aggregation [10]. We measured the aggregation of several human IgG submitted to aerosolization using a vibrating-mesh nebulizer, which is expected to be less deleterious than other nebulizers [42,43]. Flow cell microscopy and dynamic light scattering revealed that the number and size of aggregates were Ab dependent, confirming the results in the literature on the necessity of a case-by-case approach. Moreover, drastic differences were observed between commercial Abs (hIgG1-1 to 3) and murine Abs (mIgG2b-1 and -2), where the latter displayed higher aggregation upon nebulization. This may be explained by the fact that commercial Abs underwent advanced development programs and were thus selected for their limited aggregation potency. We next determined whether the aggregates found in the therapeutic product after nebulization may induce immunogenicity using an MoDC-based assay, which has been widely used to describe the potential immunogenicity of biotherapeutics or unwanted products [44]. Our analysis revealed that nebulized and aggregated hIgG1, in particular, hIgG1-1, were able to induce MoDC activation and maturation, as evidenced by the enhanced secretion in cytokines and increased expression of co-stimulatory proteins on MoDC. This response correlated with the fraction of micron-sized aggregates in the Ab aerosol. There are still discrepancies regarding the size and type of aggregates involved in the generation of immunogenic responses [45]. This depends on the type and strength of the stress applied and the multiple experimental protocols, which have been described [21,46]. Here, hIgG1-1 aerosol is mainly composed of small-sized particles (2–10 µm), which have been shown to enhance the immune response and be the most immunogenic [45,47].

One potential bias regarding Ab aggregation could be attributed to the aerosol collection step, which uses a polypropylene tube to re-pool aerosol droplets into a bulk liquid. Indeed, it has been evidenced that the aerosol collection device could influence protein stability, generating different aggregation profiles [11,48]. In this context, we used the VITROCELL technology, which was developed to improve the reliability of toxicological studies for aerosolized compounds on air–liquid interface cell cultures [49,50]. This system, which allows the direct deposition of aerosols on cells, avoids the collection step [51]. Our results showed that direct hIgG1-1 deposition on MoDCs resulted in cell activation, as evidenced by the similar increase in both cytokine production and expression of costimulatory proteins than those obtained with the nebulized-collected Ab. Thus, the collection system used here was considered relevant for these assays and was kept for the in vivo experiments.

Considering the complexity of the lung mucosal-associated immune system, it was necessary to use an animal model to predict the immunogenicity of nebulized Ab in vivo [52,53]. We chose murine antibodies to avoid a high immunogenic background response due to the non-specific activation of the mouse immune system by foreign proteins. In our study, we observed that single or repeated administration of a nebulized antibody in the lungs induced a significant contraction of the total and immune cell number in the airways starting quickly after the administration as compared to native Ab or saline controls. This was dependent on the number of aggregates as low aggregated Ab, filtered or PS80 preparation of nebulized Ab did not have any impact on cell number. Interestingly, at later time points, this contraction reached the lung parenchyma and the spleen, significantly reducing the number of both myeloid and lymphoid cells. We wondered whether this cell number reduction induced by the administration of nebulized Ab was associated with cell death. We observed that apoptosis was significantly increased in lung total cells and in leukocytes after the airway administration of nebulized Ab. Interestingly, these cellular injuries were dependent on the presence of aggregates, as Ab formulated with surfactant, known to limit aggregation [10], and even filtered nebulized Ab preparation did not promote cell death. These adverse effects were also dependent on the route of administration as nebulized Ab administered through the intravenous route was not associated with the same reduction in cell number. The complete understanding of the molecular and cellular mechanism associated with the massive cytotoxicity of nebulized and aggregated antibodies requires further investigations.

Protein aggregation underlies many chronic diseases where aggregates are thought to elicit injury, including cell apoptosis [54,55]. To the best of our knowledge, this is the first study reporting that extracellular therapeutic protein aggregates may sensitize the host to cytotoxicity. This cellular injury occurred in the absence of a pro-inflammatory response, which is contradictory with the current paradigm regarding the induction of innate immune responses by protein aggregates [13]. This discrepancy may come from the attributes of Ab aggregates associated with nebulization, as compared to those associated with other aggregation stresses, including the formation of neoepitopes, the immunomodulatory properties of the aggregates, the exposure of post-translational modifications or the generation of danger signals [56]. A complete understanding of the physical mechanisms accounting for the immunogenic properties of nebulized Ab aggregates is beyond the scope of this study. Immunogenicity may also be associated with the breakdown of self-tolerance rather than an active immune response [56]. It is particularly concordant with repeated exposure, which may occur during dosing regimens for chronic disease [36].

In conclusion, we demonstrated that aerosolization using a clinically-relevant nebulizer induced Ab aggregation and resulted in immune cell activation and immunocytotoxicity in vivo. Although there are still many questions to address to better understand the relationship between Ab aggregates and immunogenicity, our findings point to a significant role for the route of administration in the immunogenic/biological response associated to Ab aggregates. Further investigations will be required to determine the types and the number of aggregates and the role for the Fc domain in the immunocytotoxic response of Ab aggregates produced during nebulization. Our findings also highlight the importance to further explore the different methods (protein engineering, aerosolization process and formulations) to stabilize Ab during aerosolization to minimize risks for the patients.

## Figures and Tables

**Figure 1 pharmaceutics-14-00671-f001:**
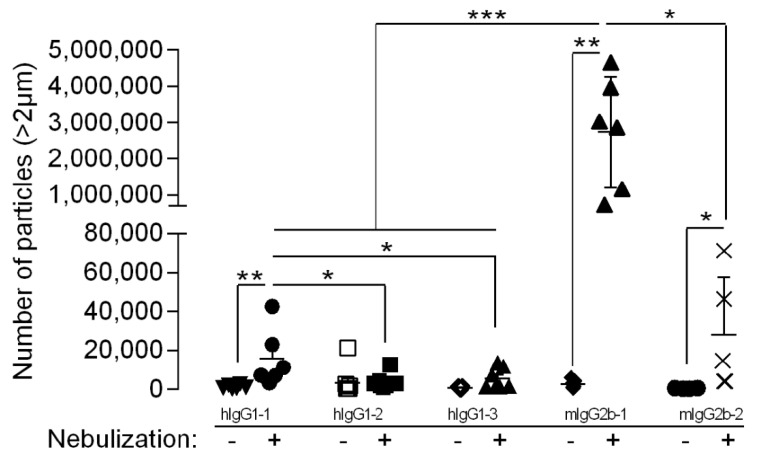
Antibodies were either nebulized using an Aeroneb Pro^TM^ vibrating-mesh nebulizer and collected (Nebulization +) or left untreated (Nebulization −). The total number of particles (with diameter >2 µm) was quantified using a flow microscope. The data are quoted as the mean ± SEM. *, **, ***: *p* < 0.05, *p* < 0.01 and *p* < 0.001, respectively, in a one-way ANOVA with Newman–Keuls correction for multiple comparisons. The results represent three to eight independent nebulizations.

**Figure 2 pharmaceutics-14-00671-f002:**
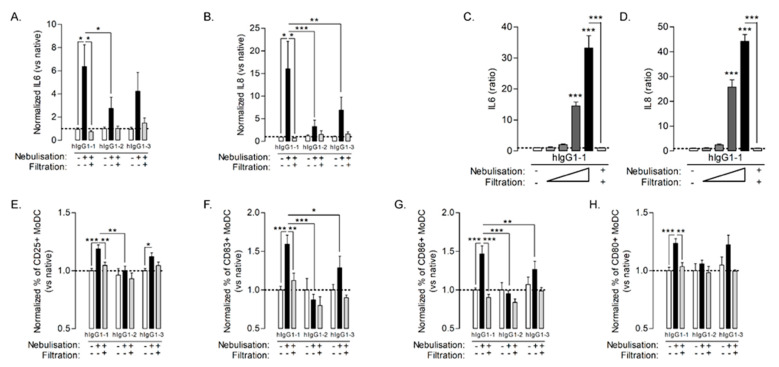
MoDC were stimulated using equal final concentration of Ab at 100 µg/mL, either native (white bars), nebulized (black bars) or nebulized and 0.45 µm-filtered (gray bars) for 18 h. IL6 (**A**) and IL8 (**B**) were quantified in cell-free supernatant. MoDC were stimulated with 1, 10, 100 or 200 µg/mL (gray to black bars) of nebulized hIgG1 or 100 µg/mL of nebulized and 0.45 µm-filtered hIgG1 (last bar) or left untreated (white bars) for 18 h. IL6 (**C**) and IL8 (**D**) were quantified in cell-free supernatant. MoDC were stimulated using equal final concentration of Ab at 100 µg/mL either native (white bars), nebulized (black bars) or nebulized and 0.45 µm-filtered (gray bars) for 18 h. CD25 (**E**), CD83 (**F**), CD86 (**G**) and CD80 (**H**) expression were measured using flow cytometry. The data are quoted as the mean ± SEM. *, **, ***: *p* < 0.05, *p* < 0.01 and *p* < 0.001, respectively, in a one-way ANOVA with Newman–Keuls correction for multiple comparisons. The results are representative of six independent experiments (*n* = 6–9 technical replicates/experiment).

**Figure 3 pharmaceutics-14-00671-f003:**
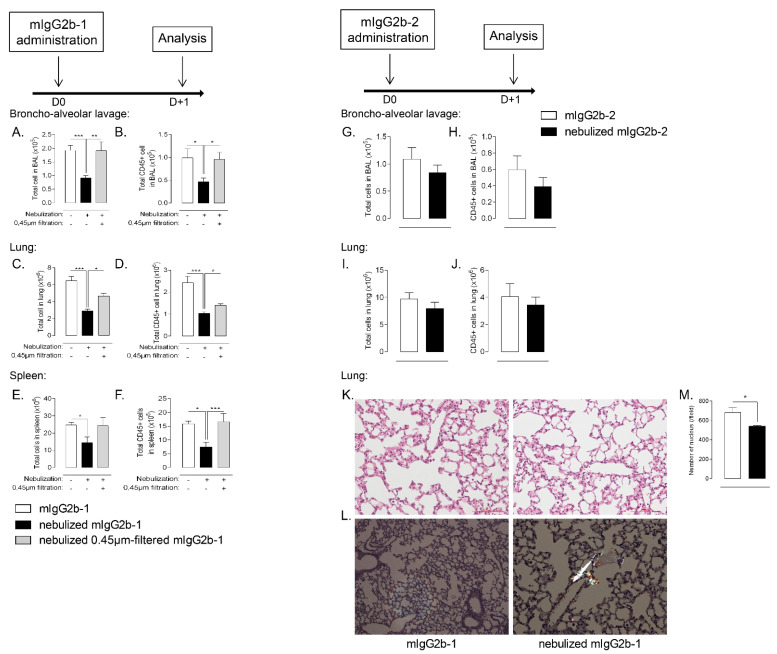
B6 mice received a 40 µL orotracheal instillation of mIgG2b-1 at 100 µg/mL either native (white bars), nebulized (black bars) or nebulized and 0.45 µm-filtered (gray bars). The total number of cells (**A**,**C**,**E**) and CD45+ cells (**B**,**D**,**F**) were quantified in BAL (**A**,**B**), in the lungs (**C**,**D**) and the spleen (**E**,**F**) using flow cytometry, 18 h after the administration. B6 mice received a 40 µL orotracheal instillation of mIgG2b-2 at 100 µg/mL either native (white bars) or nebulized (black bars). The total number of cells (**G**,**I**) and CD45+ cells (H and J) were quantified in BAL (**G**,**H**), in the lungs (**I**,**J**) using flow cytometry, 18 h after the administration. Lung tissues of mice treated with either native (white bars) or nebulized (black bars) mIgG2b-1 were histologically examined 18 h after the administration. (**K**) Hematoxylin-eosin sections were used to quantify cell nucleus (**M**) by machine-learning (see material and methods section). (**L**) Congo red sections were observed under polarized light. Aggregates are identified as apple-green birefringence artifacts. 10 sections/mouse were observed at x20 magnification and used for machine-learning quantification. The data are quoted as the mean ± SEM. *, **, ***: *p* < 0.05, *p* < 0.01 and *p* < 0.001, respectively, in a one-way ANOVA with Newman–Keuls correction for multiple comparisons. The results are representative of three independent experiments (*n* = 5 mice/experiment).

**Figure 4 pharmaceutics-14-00671-f004:**
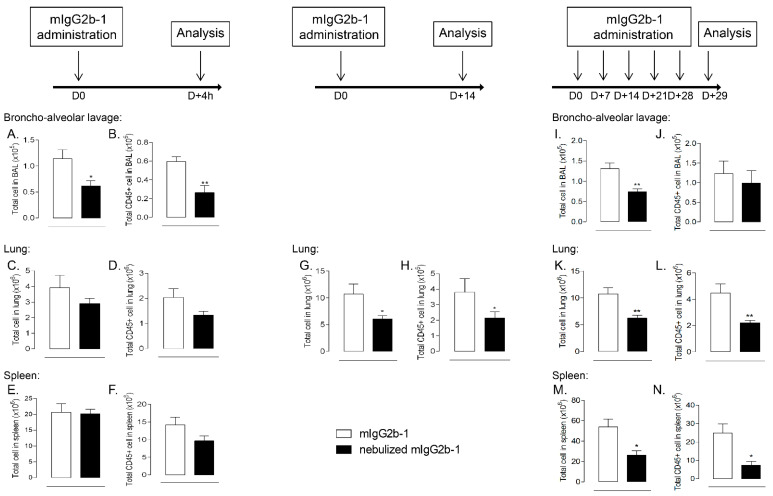
B6 mice received a 40 µL orotracheal instillation of mIg2b-1 at 100 µg/mL either native (white bars) or nebulized (black bars) through the airways at D0, or D + 0, D + 7, D + 14, D + 21 and D + 28. The total number of cells (**A**,**C**,**E**,**G**,**I**,**K**,**M**) and CD45+ cells (**B**,**D**,**F**,**H**,**J**,**L**,**N**) were quantified in BAL (**A**,**B**,**I**,**J**), in the lungs (**C**,**D**,**G**,**H**,**K**,**L**) and the spleen (**E**,**F**,**M**,**N**) using flow cytometry, 4 h, 14 days or 29 days after the first administration. The data are quoted as the mean ± SEM. *, **: *p* < 0.05 and *p* < 0.01, respectively, in a t-test. The results are representative of two independent experiments (*n* = 5 mice/experiment).

**Figure 5 pharmaceutics-14-00671-f005:**
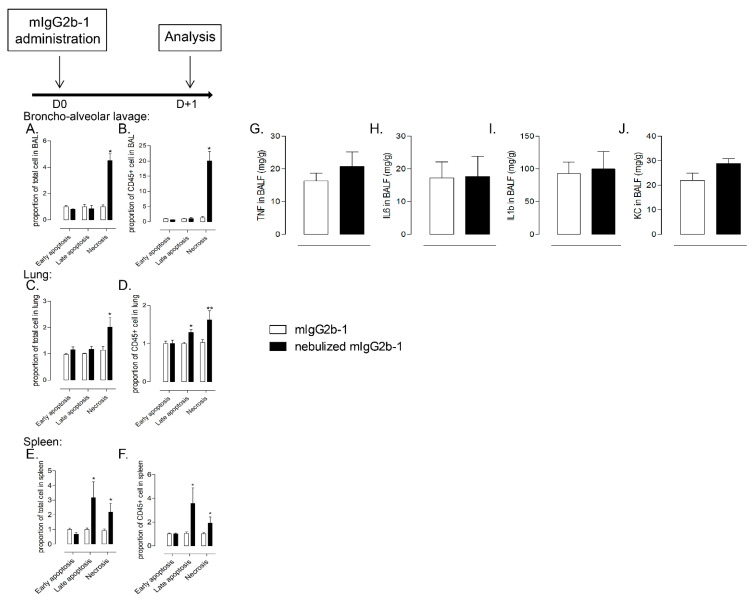
B6 mice received a single 40 µL orotracheal instillation of mIg2b-1 at 100 µg/mL either native (white bars) or nebulized (black bars). The proportion of early apoptotic cells (Annexin-V+/PI-), late apoptotic cells (Annexin-V+/PI+) and necrotic cells (Annexin-V-/PI+) were quantified in total cell (**A**,**C**,**E**) or CD45+ cell (**B**,**D**,**F**), 18 h after the administration in BAL (**A**,**B**), lungs (**C**,**D**) and spleen (**E**,**F**) relative to mice treated with native Ab using flow cytometry. The concentrations of TNF (**G**), IL6 (**H**), IL1b (**I**) and KC (**J**) in BALF were determined 18 h after the administration. The data are quoted as the mean ± SEM. *, **: *p* < 0.05 and *p* < 0.01, respectively, in a t-test. The results are representative of three independent experiments (*n* = 5 mice/experiment).

**Figure 6 pharmaceutics-14-00671-f006:**
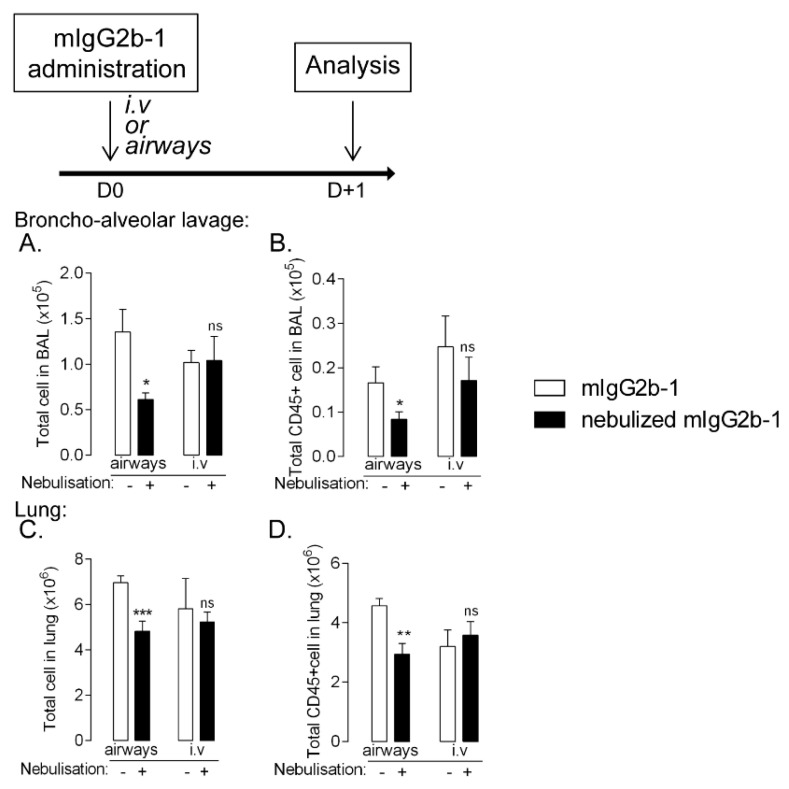
B6 mice received a 40 µL orotracheal instillation or 100 µL intravenous injection of mIgG2b-1 at 100 µg/mL either native (white bars) or nebulized (black bars). The total number of cells (**A**,**C**) and CD45+ cells (**B**,**D**) were quantified in BAL (**A**,**B**) and in the lungs (**C**,**D**) using flow cytometry, 18 h after the administration. The data are quoted as the mean ± SEM. *, **, ***: *p* < 0.05, *p* < 0.01 and *p* < 0.001, respectively, in a t-test. The results are representative of two independent experiments (*n* = 5 mice/experiment).

**Figure 7 pharmaceutics-14-00671-f007:**
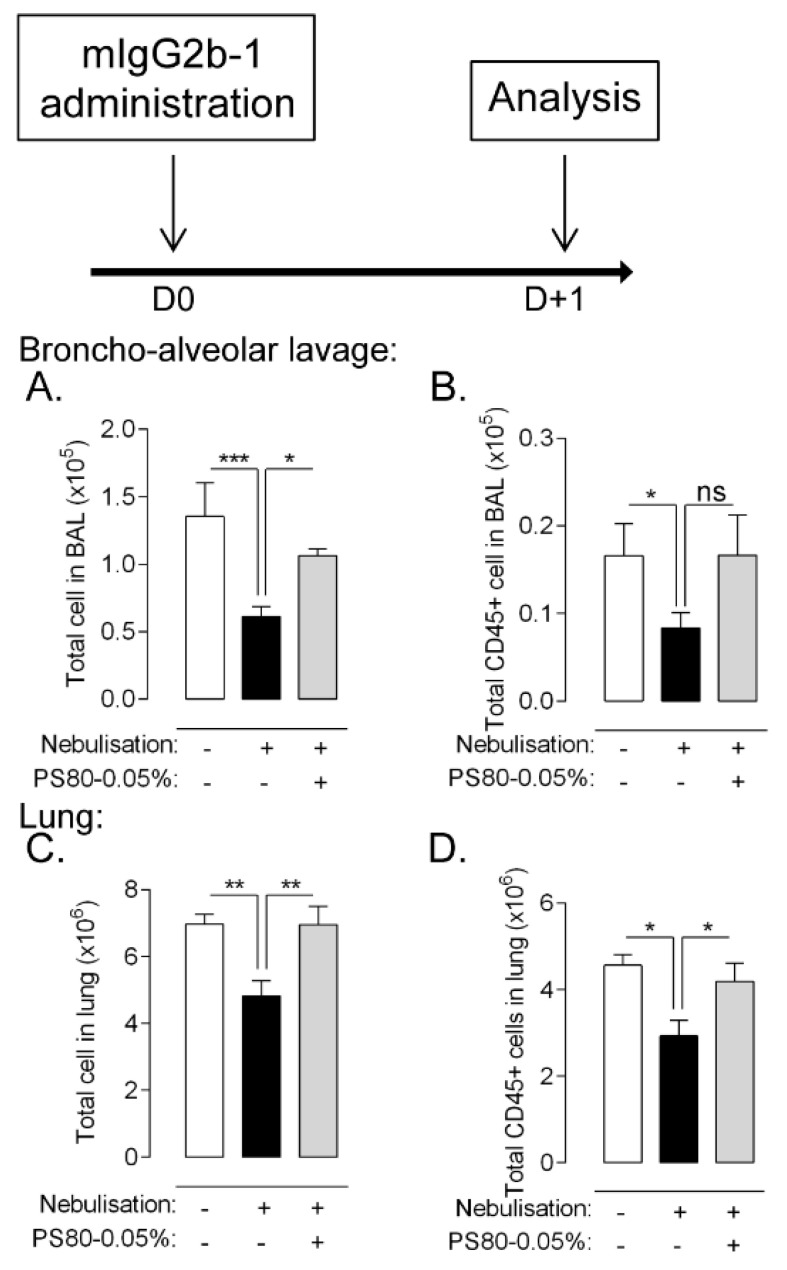
B6 mice received a 40 µL orotracheal instillation of mIgG2b-1 at 100 µg/mL either native (white bars), nebulized (black bars) or nebulized + 0.05%-PS80(gray bars). The total number of cells (**A**,**C**) and CD45+ cells (**B**,**D**) were quantified in BAL (**A**,**B**) and in the lungs (**C**,**D**) using flow cytometry, 18 h after the administration. The data are quoted as the mean ± SEM. *, **, ***: *p* < 0.05, *p* < 0.01 and *p* < 0.001, respectively, in a t-test. The results are representative of two independent experiments (*n* = 5 mice/experiment).

**Table 1 pharmaceutics-14-00671-t001:** Particle size distribution (%) of nebulized antibodies.

Antibody	2–5 µm	5–25 µm	>25 µm
hIgG1-1	84.3	14.7	1.1
hIgG1-2	78.4	21.1	0.4
hIgG1-3	53.8	45.8	0.4
mIgG2b-1	67	32.3	0.7
mIgG2b-2	57.8	40.7	1.6

## Data Availability

The data supporting this article will be shared on reasonable request to the corresponding author.

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
