# Peer review of "Aggregates Associated with Instability of Antibodies during Aerosolization Induce Adverse Immunological Effects"

_pharmaceutics, 2022, doi:10.3390/pharmaceutics14030671_

Round 1

Reviewer 1 Report

This study is well designed and presented. I have some inquiries about the work design, especially in vivo experiment; the number of mice is not mentioned in the methods, the controls, I can not see any control, especially negative control, and lastly, did the administered mice show any clinical signs? 

  • there are some punctuations in the writing as; in vitro and in vivo should be written italic
  • What is the recommendation of this study to use Ab therapy?
  • The authors used many references in the results why?
  • Samples of each parameter and the replicates are not clear
  • This title in results ( The effect of nebulization-mediated antibody aggregates on immune cell homeostasis is specific of the pulmonary route) is not shown in the methods, especially when the authors compare the lung route with the intravenous route
  •  

Author Response

This study is well designed and presented. I have some inquiries about the work design, especially in vivo experiment; the number of mice is not mentioned in the methods, the controls, I can not see any control, especially negative control, and lastly, did the administered mice show any clinical signs? 

We thank the reviewer for this comment. We have clearly indicated the number of mice/group and the number of experiments in both Methods and Figure legends. Regarding the use of a negative, as you mention in the text, we did not see any difference in term of cell count between untreated (sham) and native-mAb-treated animals. In order to limit the number of animals used in the study, we only compared native-mAb versus nebulized-mAb-treated animals.

We did not observe any sign of toxicity (body-weight loss) in mice administered with nebulized IgG both after single or repeated administrations. This has been mention in the revised version of the manuscript.

  • there are some punctuations in the writing as; in vitro and in vivo should be written italic

Corrections have been made in the revised manuscript.

  • What is the recommendation of this study to use Ab therapy?

This original article is exploratory and there is still many questions to address before being able to provide any recommendation on the inhaled Ab therapy. Although, as demonstrated by other groups and ours, direct delivery of Ab in the respiratory tract is attractive and promising, it is important – as highlighted in this study -  to take into account nebulization-associated antibody aggregation since it can rise safety concerns.

  • The authors used many references in the results why?

We included references in the result section that were related to specific methodological issues or to introduce concept to help the reader.

  • Samples of each parameter and the replicates are not clear

Thank you for this comment. We modified the manuscript to introduce more clearly the replicates/n for the different experiments.

For Figure 1: the number of aggregates are represented as a mean of 3 to 8 independent mAb nebulization.

For Figure 2: we have pooled 6 experiments including independent MoDC culture/differentiation, Ab nebulization, cell stimulation, flow cytometry analysis/ELISA. For each independent experiments, 6-9 technical replicates have been performed.

For in vivo analysis: each experiment was done using 5 mice per group and 2-3 independent experiments were performed for the figures. Accordingly, the results correspond to a total of 10-15 mice.

  • This title in results ( The effect of nebulization-mediated antibody aggregates on immune cell homeostasis is specific of the pulmonary route) is not shown in the methods, especially when the authors compare the lung route with the intravenous route

Thank you for this comment. This paragraph depicted the effect of nebulized-associated antibody aggregates according to their route of administration and our results are supported by Figure 6 and Supplementary figure 6. We added a brief description on the different routes of administration in the Material & Methods section.

Reviewer 2 Report

Thank you for the opportunity to review this paper investigating adverse immunological effects induced by aggregates associated with instability of antibodies during aerosolization entitled “Aggregates associated with instability of antibodies during aer- 2 osolization induce adverse immunological effects” by Secher et al. This manuscript looks extensively at the impact on the immune system from aggregates that result from antibody instability during aerosolization after delivery by inhalation. Since therapeutic antibodies can encounter numerous stresses during various phases before and after antibody bioprocessing, there are higher risks for aggregation that could impose even more significant risks for patients. While there is literature that shows some of these stressed conditions, it is not representative of commercial products as the stressors shown in current literature do not encompass all situations that occur from product development through clinical use. The author has used in-vivo and in-vitro models to show that the aggregation resulting from therapeutic antibody nebulization induced immune cell overactivation. Overall, their results showed antibody heterogeneity and confirmation of antibody aggregation during nebulization. In addition, the inhalation of aggregated IgG locally and systemically impacted cellular homeostasis over time, inducing cell death. In addition, data suggest that the specific route of administration, inhalation, played a critical role in these adverse effects seen, and limiting AB aggregation in nebulization might alleviate the harmful impact on cell homeostasis. So, while this study provides support to the hypothesis that protein aggregation underlies chronic diseases and that therapeutic protein aggregates may sensitize the host to say toxicity, there are a few discrepancies that can be further elaborated on or addressed in future directions. Overall, this was a very well-written and engaging paper.

Minor Concerns:

  • There are minor subject-verb
  • agreement errors in some sentences.
  • The whole manuscript needs to revise related to typos, such as C57/BL6 to C57BL/6, etc. In addition, with the multiple time abbreviation for the same word.
  • If the artifact in Figure 3, L, is aggregated antibodies, can you address this in the figure description?
  • Lung tissue section staining is not well. I would suggest replacing all these representative images with another with good graphics.
  • Please revise Figure 1 bar graph to scatter-bar graph. It will help researchers to see individual differences.
  • I am unable to see Figures 4 E and F could be the formatting issue. Please revise it accordingly.
  • Some figures are represented non-significance by “ns” such as Figures 6 and 7, while others are not. Please edit in figure legend if you are using any symbol or letter. 
  • While it might be outside the current scope of your project, I think it might be beneficial within your conclusions to elude towards some future direction. I would consider focusing on continued efforts to either investigate other potential mechanisms of immunogenicity in these aggregated proteins or methods of improvement/ stabilization of therapeutic antibodies That would improve efficacy with minimal risk to patients.

Author Response

Minor Concerns:

  • There are minor subject-verb agreement errors in some sentences.

Our manuscript was entirely reviewed by a native English-speaker (RML) in order to correct these issues.

  • The whole manuscript needs to revise related to typos, such as C57/BL6 to C57BL/6, etc. In addition, with the multiple time abbreviation for the same word.

Typos have been corrected and wordings homogenized along the manuscript. “Hours” is written in full letter in the text and as “h” abbreviation in the figure legends.

  • If the artifact in Figure 3, L, is aggregated antibodies, can you address this in the figure description?

We do believe that this Congo-Red artifact which gave a green birefringence signal under polarized light is a mAb aggregates. We have described it in the figure legend.

  • Lung tissue section staining is not well. I would suggest replacing all these representative images with another with good graphics.

Figure 3K depicted representative images of HE-stained lung section on which we have performed a machine-learning analysis to count cell nucleus, in order to support our flow cytometry analysis. The result of this analysis is depicted in Figure 3M

  • Please revise Figure 1 bar graph to scatter-bar graph. It will help researchers to see individual differences.

This has been modified in the revised version of the manuscript

  • I am unable to see Figures 4 E and F could be the formatting issue. Please revise it accordingly.

Figure 4 has been provided entirely in the submitted manuscript. We apologize for this formatting issue, which is not beyond our control and hope that Figures 4 E and F will be readable in the revised version of the manuscript

  • Some figures are represented non-significance by “ns” such as Figures 6 and 7, while others are not. Please edit in figure legend if you are using any symbol or letter. 

We have modified figure legend accordingly.

  • While it might be outside the current scope of your project, I think it might be beneficial within your conclusions to elude towards some future direction. I would consider focusing on continued efforts to either investigate other potential mechanisms of immunogenicity in these aggregated proteins or methods of improvement/ stabilization of therapeutic antibodies That would improve efficacy with minimal risk to patients.

We thank the reviewer for this useful comment. We have now commented this points in the discussion section.

Reviewer 3 Report

The introduction part is lack background, aim, and hypothesis.

The goal is not clear in the discussion or conclusion.

The experimental protocols are lack quality evidence.

What is the novelty of this study?

Why do we need to have this research published in scientific communities? 

Author Response

The introduction part is lack background, aim, and hypothesis.

The goal is not clear in the discussion or conclusion.

The experimental protocols are lack quality evidence.

What is the novelty of this study?

Why do we need to have this research published in scientific communities? 

The reviewer may not be familiar with the field of “aerosol medicine” and the challenges associated to inhalation of protein therapeutics. Inhalation and antibody recovered around 7000 hits in PUBMED in the past 2 years and several molecules, either targeting the nose or the lungs, have reached clinical trials in the past decade. With the SARS-CoV-2 pandemic, there has been an increasing focus of interest on this delivery route, which matches the pathogen entrance and replication site. Thus, we believe that this study will interest the scientific and pharma communities developing such approach.

It is noteworthy that despite promising preclinical studies, none inhaled Ab has obtained market approval, highlighting the challenges/hurdles that remain to overcome for their successful development. We have been working on this topic over the past 15 years, with a step by step approach, to decipher the pharmacodynamics, the pharmacokinetics, the safety of the inhalation route for therapeutic antibodies. As referred in the introduction, one of the main challenges associated to inhalation delivery of antibody is their stability during the aerosolization process. Here, we demonstrated that Ab instability during aerosolization has major biological consequences, which are related both to the aggregates and their route of administration, throught the airways.

Although there are still issues to address to understand better the relationship between Ab aggregates and immunogenicity after inhalation, our findings point the importance to explore the impact of antibody instability and methods (protein engineering, aerosolization process, formulations,…) to stabilize further Ab during aerosolization to minimize risks for the patients.

Round 2

Reviewer 3 Report

This reviewer is glad to have such wonderful and informative responses from the corresponding author. However, this reviewer is regrettably rejecting this manuscript because the explanation was not satisfactory; it was more like historical story rather than scientific things. It's not about previous citations or how many years you are working on, its all about what are you going to put on the table. This reviewer also believes that any form of this manuscript would not be sufficient to accept in this stage.

Thank you.